# EXPLORING CONTEXTUAL MODELING WITH LINEAR COMPLEXITY FOR POINT CLOUD SEGMENTATION

## ABSTRACT

Point cloud segmentation is an important topic in 3D understanding that has traditionally has been tackled using either the CNN or Transformer. Recently, Mamba has emerged as a promising alternative, offering efficient long-range contextual modeling capabilities without the quadratic complexity associated with Transformer's attention mechanisms. However, despite Mamba's potential, early efforts have all failed to achieve better performance than the best CNN-based and Transformer-based methods. In this work, we address this challenge by identifying the key components of an effective and efficient point cloud segmentation architecture. Specifically, we show that: 1) Spatial locality and robust contextual understanding are critical for strong performance, and 2) Mamba features linear computational complexity, offering superior data and inference efficiency compared to Transformers, while still being capable of delivering strong contextual understanding. Additionally, we further enhance the standard Mamba specifically for point cloud segmentation by identifying its two key shortcomings. First, the enforced causality in the original Mamba is unsuitable for processing point clouds that have no such dependencies. Second, its unidirectional scanning strategy imposes a directional bias, hampering its ability to capture the full context of unordered point clouds in a single pass. To address these issues, we carefully remove the causal convolutions and introduce a novel Bidirectional Strided SSM to enhance the model's capability to capture spatial relationships. Our efforts culminate in a novel architecture named MEEPO that effectively integrates the strengths of CNN and Mamba. MEEPO surpasses the previous state-of-the-art method, PTv3, by up to **+0.8** mIoU on multiple key benchmark datasets, while being **42.1%** faster and **5.53×** more memory efficient. Our code will be released.

## 1 INTRODUCTION

Point cloud segmentation is an important topic in 3D understanding that has gained significant attention from the research community in recent years. Numerous neural network architectures have been proposed for this task, with CNN-based and Transformer-based designs being the two most prominent. Currently, Transformer-based methods consistently deliver the highest performance across numerous benchmarks. Their success is frequently credited to the attention mechanism, which can adequately capture and model context. However, the quadratic complexity of self-attention in Transformers poses a significant challenge for point cloud processing, especially when handling a large number of points. To address this, researchers have explored more efficient strategies, such as aggressive downsampling (Zhao et al., 2021; Wu et al., 2022; Pan et al., 2021), efficient attention algorithms (Yang et al., 2023), and windowed attention mechanisms (Wang, 2023; Lai et al., 2022; Wu et al., 2024). While these methods help reduce computational costs, they achieve this at the expense of valuable spatial and geometric information, potentially weakening the Transformer's modeling capability and hindering its contextual understanding of the point cloud (Shen et al., 2021).

Recently, State Space Models (SSMs) like Mamba (Lieber et al., 2024; Gu & Dao, 2024; Pióro et al., 2024; Wang et al., 2024b) have emerged as a promising alternative to Transformers. These models combine aspects of recurrent neural networks (Cho et al., 2014; Hochreiter & Schmidhuber, 1997) and convolutional neural networks (LeCun et al., 1989) within a framework rooted in classical state space theory. Similar to Transformers, SSMs provide robust contextual modeling capabilities. However, unlike Transformers, which scale quadratically with sequence length, SSMs scale linearly

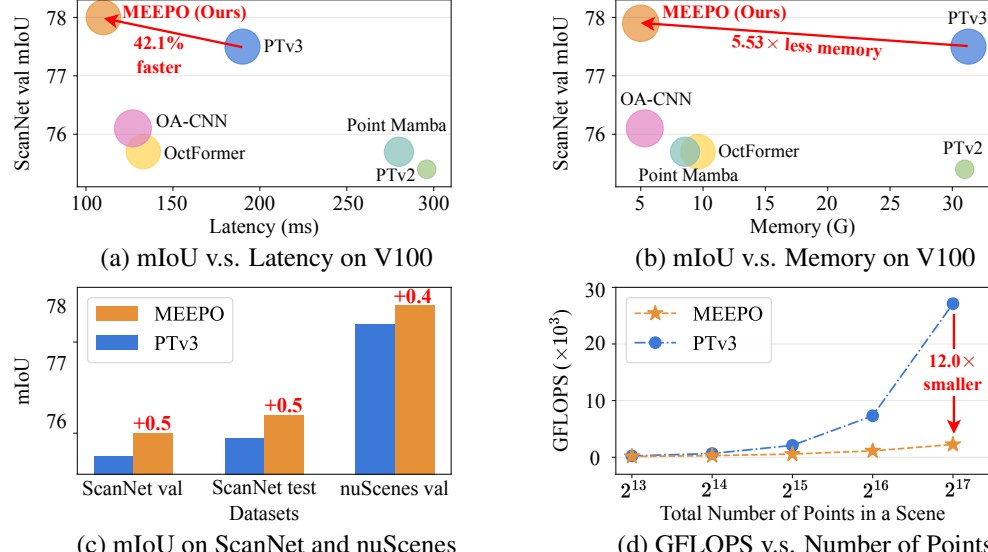

Figure 1: Performance and efficiency comparisons between our proposed method, MEEPO, and other leading segmentation networks using the ScanNet and nuScenes dataset. By carefully combining the strengths of existing architectures, MEEPO surpasses all previous leading methods while using much lower latency and memory. Furthermore, it easily scales to scenes with a much larger number of points and progressively improves performance as the input sequence length increases.

and maintain constant memory usage during inference. This efficiency makes SSMs particularly advantageous for tasks that require handling extensive contexts.

Although numerous early attempts have been made to utilize SSMs for point cloud segmentation (Liang et al., 2024b; Zhang et al., 2024; Liu et al., 2024b), these efforts have been largely *discouraging*, as their performance significantly lags behind that of leading CNN-based and Transformer-based models. As illustrated in Tab. 1, the recently proposed Point

Table 1: Performance and Latency Comparison among representative Mamba-based, CNN-based and Transformer-based networks on ScanNet.

| case | mIoU↑ | latency↓ |
|---|---|---|
| Point Mamba (Liu et al., 2024b) | 75.7 | 280 |
| OA-CNN (Peng et al., 2024) | 76.1 | 141 |
| PTv3 (Wu et al., 2024) (current best) | 77.5 | 183 |

Mamba network (Liu et al., 2024b) not only incurs roughly twice the latency of the CNN-based OA-CNN (Peng et al., 2024) and Transformer-based PTv3 (Wu et al., 2024) but also underperforms them by **0.4** points and **2.2** points in mIoU on the ScanNet (Dai et al., 2017) validation dataset, respectively. These shortcomings underscore the ongoing debate regarding the optimal architecture for point cloud segmentation, leading to the pivotal question: *What constitutes an efficient and effective model architecture for point cloud segmentation?*

In this work, we aim to provide valuable insights for the design of point cloud segmentation architectures. To achieve this, we conduct a preliminary analysis in Sec. 3 to examine the properties of the three most popular architectures, namely the CNN-based, Transformer-based and Mamba-based networks. Using the meta-architecture presented in Fig. 2, which seamlessly integrates different operators from the CNN, Transformer and Mamba architectures, we compare these architectures across three key dimensions: *contextual understanding capability*, *local sensitivity*, and *network efficiency*. Our analysis reveals that while CNNs excel at local modeling, they lack the ability to capture broader context. Transformers can adequately capture contextual information but are inefficient due to unnecessary long-range attention and quadratic computational complexity. Mambas strike a balance by efficiently providing essential contextual understanding with linear complexity. Given the distinct strengths and limitations of each architecture, we hypothesize that an integrated architecture combining their best features could potentially yield a more powerful and efficient model.

Building on the insights gained, we systematically explore various block choices, placements, and quantities within the previously proposed meta-architecture to determine the optimal arrangement. Through this process, we identify the CNN-Mamba block as the optimal elementary block for our architecture. As depicted in Fig. 6(a), the CNN-Mamba block comprises a sequential stack of sparse

convolution layers and Mamba modules. Notably, the Attention module is ultimately excluded from the final architecture because the more efficient Mamba module can already provide sufficient contextual understanding for this task.

With the macro-level design established, we turn our attention to the micro-level design, specifically assessing whether the standard Mamba module, originally designed for sequential processing, is suitable for point cloud segmentation. Our investigation shows that it is not. In particular, we identify two major shortcomings of the standard Mamba module when applied to this task:

1. **Loss of spatial information due to enforced causality**: Mamba's use of causal convolutions introduces unnecessary artificial dependencies that can disrupt the inherent spatial relationships in point cloud data, ultimately reducing its effectiveness for point cloud data.

2. **Directional bias due to unidirectional scan**: Mamba's unidirectional scan strategy inherently favors certain data points over others, creating a directional bias. This bias undermines its ability to fully grasp the context of unordered point clouds in a single pass.

To address these issues, we propose two corresponding improvements to the standard Mamba module. Specifically, we 1) remove the causality constraint, and 2) incorporate a novel *Bidirectional Strided SSM* to enhance its context and spatial understanding.

Our work culminates in MEEPO, a novel point cloud segmentation architecture which seamlessly integrates Mamba's efficiency and strong contextual understanding with the local sensitivity of CNN. As shown in Fig. 1, MEEPO not only achieves significantly lower inference latency but also surpasses the performance of other leading segmentation models across both indoor and outdoor scenes. Specifically, it outperforms the previous best method, PTv3, by up to **+0.8** mIoU across ScanNet, ScanNet200, S3DIS, and nuScenes datasets, with **42.1%** smaller latency and **5.53×** smaller memory usage. MEEPO also scales much better with respect to the size of point cloud. As shown in Fig. 1(d), it achieves up to $12\times$ fewer GFLOPs than PTv3 when processing a scene with $2^{17}$ $(131,072)$ points.

In summary, the contributions of our paper are as follows:

1. We carefully analyze existing point cloud segmentation networks, identifying the importance of *spatial locality* and *robust contextual understanding* in achieving high performance. We then reveal that Transformer's global attention is unnecessary and inefficient for this task as Mambas offer comparable contextual modeling with linear complexity.

2. We propose two corresponding solutions to address the limitations of Mamba when applied to point cloud segmentation, namely the removal of the causality constraint and the incorporation of an innovative *Bidirectional Strided SSM* to enhance contextual understanding.

3. We introduce MEEPO, a novel architecture that solely utilizes the efficient sparse convolution to provide spatial locality and the efficient SSM to provide robust contextual understanding. MEEPO not only consistently outperforms previous best method, PTv3 across multiple key benchmark datasets, but is also much faster and much more memory efficient.

## 2 PRELIMINARIES

In this section, we briefly introduce state space model (SSM) to facilitate subsequent discussion.

**SSM's formulation.** SSM is a type of sequence model (Gu & Dao, 2024) that maps an input sequence $x(t) \in \mathbb{R}$ to an output sequence $y(t) \in \mathbb{R}$ through a latent state $h(t) \in \mathbb{R}^N$:

$$h'(t) = \mathbf{A}h(t) + \mathbf{B}\mathbf{x}(t), \quad y(t) = \mathbf{C}h(t). \tag{1}$$

Using zero-order hold (ZOH) rule, we can discretize the continuous parameters $(\mathbf{\Delta}, \mathbf{A}, \mathbf{B})$ as:

$$\bar{\mathbf{A}} = \exp(\Delta\mathbf{A}), \quad \bar{\mathbf{B}} = (\Delta\mathbf{A})^{-1}(\exp(\Delta\mathbf{A}) - \mathbf{I}) \cdot \Delta\mathbf{B}, \tag{2}$$

where $\Delta$ is the discretization step size and $\mathbf{I}$ is the identity matrix. Under this discretization rule, the hidden state $h_t$ can be computed efficiently as a linear recurrence:

$$h_t = \bar{\mathbf{A}}h_{t-1} + \bar{\mathbf{B}}x_t, \quad y_t = \mathbf{C}h_t. \tag{3}$$

Mamba is a recent popular SSM model that sets the SSM parameters to be functions of the input:

$$\mathbf{B}_t = f_B(x(t)) \quad \mathbf{C}_t = f_C(x(t)) \quad \mathbf{\Delta}_t = f_\Delta(x(t)) \tag{4}$$

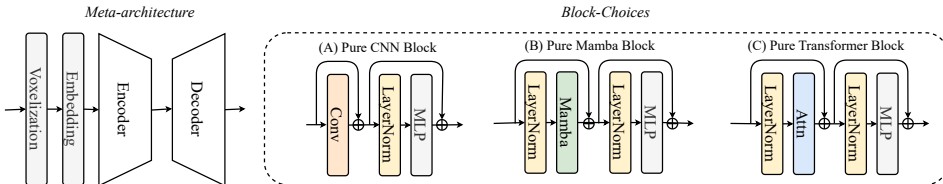

Figure 2: Proposed meta-architecture and various block options used for analysis. The model that exclusively uses choice A is called *Pure CNN*, the model that exclusively uses choice B is called *Pure Mamba*, and the model that exclusively uses choice C is called *Pure Transformer*.

This allows the model to selectively propagate or discard information based on the input. In practice, matrix $\mathbf{A}$ is typically set as a diagonal matrix, ensuring that all elements of $\bar{\mathbf{A}} = \exp(\Delta \mathbf{A})$ lie between 0 and 1. Consequently, $\bar{\mathbf{A}}$ can be viewed as a *forget gate*, which controls how much information from the previous hidden state, $h_t$, is retained (Han et al., 2024).

## 3 ANALYSIS

In this section, we delve into the key requirements for effective point cloud segmentation, showing the importance of capturing both local and contextual features via model architecture. Robust local modeling is essential for maintaining point-level consistency, especially when object boundaries are unclear, while strong contextual modeling is crucial for identifying occluded or ambiguously shaped objects. With these needs in mind, we evaluate various architectures, assessing their strengths and weaknesses in providing these important properties for effective point cloud segmentation.

**Meta-architecture for analysis.** To facilitate our analysis, we first propose a meta-architecture for point cloud segmentation. This meta-architecture follows an encoder-decoder framework, featuring a 5-stage encoder and a 4-stage decoder. Following PTv3 Wu et al. (2024), the input points are first voxelized into non-overlapping segments and arranged into an ordered sequence using alternating Z-order and Hilbert space-filling curves (Morton, 1966; Peano, 1890). These voxels are then processed by an embedding module that employs a single submanifold sparse convolution layer (Graham et al., 2018). GridPooling and GridUnpooling operations are applied at the end and beginning of each encoding stage, respectively, to downsample and upsample the point cloud (Wu et al., 2024). Within this meta-architecture, we train three similarly sized models, each using either the CNN, Mamba, or Transformer blocks, as shown in Fig. 2, to evaluate their performance both qualitatively and quantitatively. To account for the different parameter densities of these blocks, their channel sizes are adjusted accordingly to ensure similar parameter counts. These models are referred to as *Pure CNN*, *Pure Mamba*, and *Pure Transformer*, respectively. We hypothesize that the global attention mechanism of the *Pure Transformer* and the long-range sequential modeling capabilities of the *Pure Mamba* are highly effective for contextual modeling. In particular, *Pure Mamba* offers additional advantages, including enhanced inference efficiency due to its linear computational complexity and improved data efficiency enabled by the inductive bias of the forget gate. On the other hand, *Pure CNN* excels in local modeling by leveraging its inherent spatial locality.

**1. Are both local and contextual modeling important for point cloud segmentation?** To assess the individual contributions of local and contextual modeling in point cloud segmentation, we analyze the performance impact of removing key components from the current leading network, PTv3 (Wu et al., 2024). Specifically, we investigate the effects of eliminating the sparse convolution layers, which capture

Table 2: PTv3 without spatial locality perform much worse on ScanNet val.

| Case | mIoU↑ |
| --- | --- |
| PTv3 (Wu et al., 2024) | 77.5 |
|    Remove spatial locality | 69.3 (**-8.2**) |
|    Remove contextual modeling | 73.2 (**-4.3**) |

spatial locality, and attention modules, which model contextual relationships. As detailed in Tab. 2, removing the sparse convolution layers results in a substantial performance drop of **-8.2** mIoU, while removing the attention modules leads to a decrease of **-4.3** mIoU. These results highlight the critical importance of both local and contextual modeling in achieving effective point cloud segmentation.

**2. Which model is more effective at providing contextual understanding?** To investigate this, we perform another experiment using PTv3, by modifying its full attention mechanism to a window-based approach. By varying the window size, we control the amount of context the model processes.

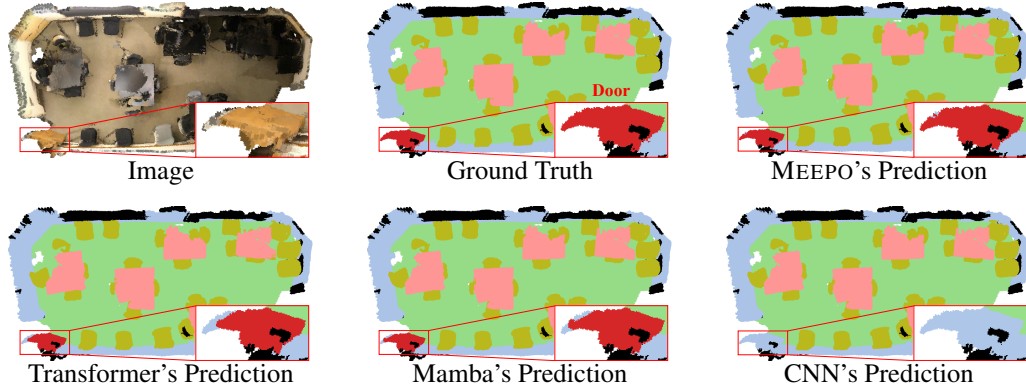

Figure 3: Both Transformer and Mamba models incorporate mechanisms to learn long-range dependencies, allowing them to accurately interpret occluded regions and areas with similar textures.

As shown in Fig. 4, increasing the window size progressively improves performance from 24 to 1024, peaking at 1024 before gradually declining as the window size grows further. Since point cloud segmentation suffers from data scarcity, the poor performance at larger window size is likely due to insufficient training data, as attention mechanisms typically require large amounts of data to be trained effectively (Dosovitskiy et al., 2021). Meanwhile, Mamba emerges as a strong candidate for modeling context due to its emphasis on local processing while still being able able to capture broader context when necessary. To provide evidence for this, we visualize the per-

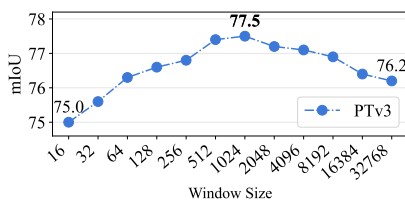

Figure 4: Analysis of a representative Transformer-based model, PTv3, demonstrates that additional context beyond a certain amount is unnecessary.

formance of different models using a representative example in Fig. 3, which depicts an office room with a challenging-to-discern door in the bottom left corner. Try to focus on the boxed region in each image and identify the object category. This task is challenging because the door's point cloud is partially occluded and its texture and color closely match the surrounding wall and floor, requiring a comprehensive understanding of the scene. As depicted in the figure, Mamba's balance of local and global context allows it to outperform the Transformer model, which in turn surpasses the CNN, thereby confirming Mamba's efficacy in contextual understanding.

**3. Which model is more effective at providing spatial locality?** To compare the performance of the aforementioned model architectures in this regard, we visualize their predictions in Fig. 5 using another insightful example of a room with a centrally placed table whose boundaries are poorly defined due to severe overexposure. Without zooming in, the boundaries are hard to discern, requiring strong local feature extraction capabilities. In this challenging scenario, the results support our hypothesis regarding spatial locality. The CNN, equipped with sparse convolutional layers specifically designed to capture local spatial patterns, outperforms both Mamba and the Transformer. Mamba exhibits slight spillage of floor pixels onto the table, while the Transformer mislabels a significant portion of the table area. These results demonstrate the CNN's superior capability in local modeling.

**4. Which model architecture is more efficient?** The computational complexities of the core operations in the Transformer, Mamba, and CNN models are as follows:

$$\Omega(\text{Transformer}) = \underbrace{4 \cdot L \cdot C^2}_{\text{qkv and output projections}} + \underbrace{2 \cdot L^2 \cdot C}_{\text{attention}} = O(L^2), \quad (5)$$

$$\Omega(\text{Mamba}) = \underbrace{9 \cdot L \cdot C \cdot N}_{\text{SSM}} + \underbrace{L \cdot C \cdot K}_{\text{depthwise conv1d}} + \underbrace{3 \cdot L \cdot C^2 \cdot E}_{\text{input and output projections}} = O(L), \quad (6)$$

$$\Omega(\text{CNN}) = \underbrace{2 \cdot C_{\text{in}} \cdot C_{\text{out}} \cdot k^3 \cdot L}_{\text{convolution}} + \underbrace{L \cdot C_{\text{in}} \cdot C_{\text{out}}}_{\text{bias addition}} = O(L), \quad (7)$$

where $C$, $C_{\text{in}}$, and $C_{\text{out}}$ represent the channel sizes, $N$ the SSM's state dimension, $E$ the SSM's expansion factor, $L$ the number of points or input sequence length, $K$ the depthwise convolution kernel size in Mamba and $k$ the convolution kernel size in sparse convolutions within CNN. Both Mamba models and CNNs scale linearly with respect to $L$, while Transformers scale quadratically.

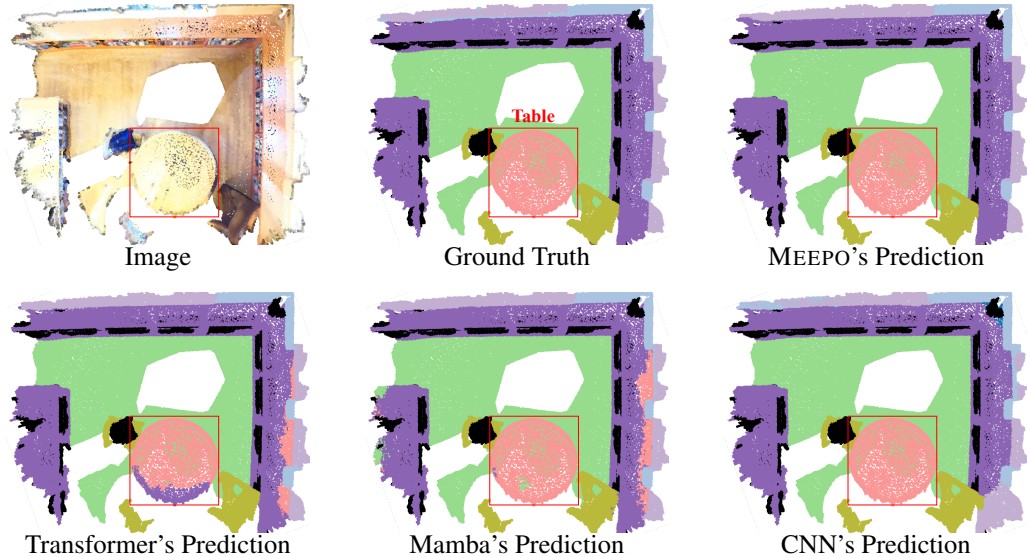

Figure 5: Comparison of model performance on tasks requiring robust local modeling. CNNs excel due to spatial convolutions, and Mambas benefit from its locally-biased forget gate. Transformers, lacking specialized local modeling mechanisms, often produce inaccurate predictions.

Consequently, for point cloud processing tasks involving hundreds of thousands of points, Transformers can be significantly slower. When comparing CNNs and Mamba models of similar sizes, CNNs are generally faster in practice. This is because CNNs typically have fewer layers when the total number of parameters is the same, as each 3D sparse convolution layer in a CNN contains significantly more parameters than a corresponding block in a Mamba model.

**Key insights:** Point cloud segmentation requires both effective local modeling and a comprehensive understanding of contextual information. While CNNs excel at local modeling, contextual modeling demands a different approach. Due to the current scarcity of large-scale point cloud segmentation datasets, Transformers cannot fully leverage their attention mechanism potential, with performance peaking at a window size of 1024. Mambas, however, strike a balance between local and global modeling, providing the necessary contextual understanding with linear complexity, without the quadratic complexity of Transformers. Given the distinct strengths and limitations of each architecture, it is crucial to explore models that can holistically integrate these capabilities.

## 4 METHOD

Building on insights from previous analysis, we introduce MEEPO, a novel model that is both efficient and effective for point cloud segmentation. MEEPO adopts the same meta-architecture presented in Fig. 2 but incorporates the CNN-Mamba block as a core component to facilitate the seamless integration of local and contextual modeling. Additionally, we introduce two micro-level modifications to the standard Mamba module to address its limitations in point cloud segmentation.

### 4.1 MACRO ARCHITECTURE DESIGN

**Optimizing block choices for point cloud segmentation.** To identify the most effective block configuration for integrating local and contextual modeling, we draw inspiration from the widely adopted sequential combination of local convolutional layers and contextual operators in previous point cloud segmentation studies (Wang, 2023; Wu et al., 2024) and evaluate two possible candidates: the CNN-Mamba block and the CNN-Transformer block. As depicted in Fig. 6, the CNN-Mamba block comprises a sparse convolution layer, followed by a Mamba module and an MLP layer. In contrast, the CNN-Transformer block follows the same structure but substitutes the Mamba module with an Attention module. Our comprehensive ablation study, presented in Tab. 7(b), demonstrates that replacing CNN-Transformer blocks with CNN-Mamba blocks throughout the network (22 blocks in total) achieves the best performance and efficiency. These findings validate our hypothesis that the Mamba module offers superior contextual understanding while maintaining efficiency.

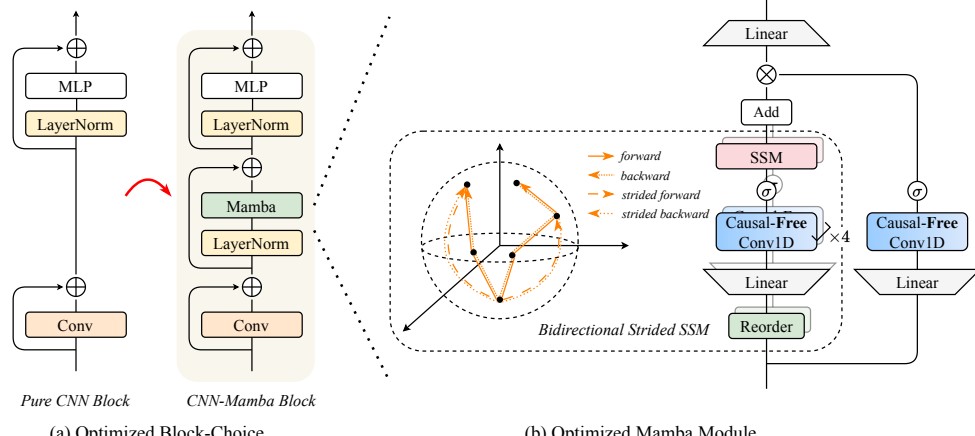

*Pure CNN Block*    *CNN-Mamba Block*

(a) Optimized Block-Choice                    (b) Optimized Mamba Module

Figure 6: Our proposed architecture, MEEPO, integrates CNN-Mamba blocks throughout the proposed meta-architecture to harness their strengths in local and contextual modeling. To optimize for point cloud segmentation, MEEPO modifies the standard Mamba by replacing causal convolutions with regular convolutions, preserving critical spatial information. Additionally, it introduces a novel *Bidirectional Strided SSM*, which enhances contextual modeling by minimizing directional bias.

## 4.2 MICRO ARCHITECTURE DESIGN

**Optimizing Mamba for point cloud segmentation.** Despite its impressive speed and performance, MEEPO without micro-level optimizations still does not surpass the leading point cloud segmentation network, PTv3, in accuracy. This aligns with previous research, which consistently demonstrates that attempts to apply Mamba have all failed to outperform well-established models for this task (Zhang et al., 2024; Liu et al., 2024b). A closer analysis reveals that the standard Mamba, originally designed for sequential data, is inherently unsuited for processing unordered point clouds. We identify two key limitations in the standard Mamba and propose the solutions to overcome them.

**1. Loss of spatial information due to enforced causality:** Mamba was originally designed for sequential data with clear causal relationships. However, point clouds lack such dependencies, as their spatial relationships are multidimensional, requiring simultaneous consideration of points holistically. The causal convolutions in standard Mamba impose a causal dependency that disrupts these essential spatial relationships, making it ineffective for handling spatial data like point clouds.

*Proposed Solution: Causal-Free Mamba.* To address this issue, we propose replacing the causal convolution with a standard convolution, resulting in the *Causal-Free Mamba* module, as illustrated in Fig.6(b). This modification eliminates the limitations of the original Mamba architecture when processing spatial data, greatly enhancing its performance in point cloud segmentation tasks.

**2. Directional bias due to unidirectional scan:** Mamba's unidirectional scanning method introduces a directional bias in the representation learning of point clouds because some parts are scanned first while others are scanned later. Such sequential processing is ill-suited for orderless point cloud data, where all information should be treated equally. This approach can lead to models prioritizing information from later stages of the scan, overlooking details captured earlier. The bias is further amplified by factors such as noise, occlusions, or reduced data density at the beginning of the scan, which can degrade the quality of the early data. As a result, important features captured early in the scan may be missed or inaccurately interpreted, leading to segmentation errors where boundaries are poorly defined, and features are incorrectly merged or omitted.

*Proposed Solution: Bidirectional Strided SSM.* To address this issue, we propose an innovative multi-directional scanning approach to reduce the directional bias of Mamba. Unlike the standard Mamba, which employs a unidirectional scan, this scanning method processes data in four distinct scanning directions: forward, backward, $n$-strided forward, and $n$-strided backward. In an $n$-strided forward scan, every $n$-th token is skipped, and the scan pattern restarts from the beginning once the end is reached. For example, given the sequence 1, 2, 3, 4, 5, 6, a 2-strided forward scan would process it as 1, 3, 5, 2, 4, 6. The backward scan operates similarly but in reverse order. This multi-directional scanning approach can effectively expand Mamba's receptive field, reduce information loss, and improve overall performance by shortening the information flow path.

Table 3: Indoor semantic segmentation comparison on ScanNet, ScanNet200, S3DIS Area 5.

| Method | ScanNet | | ScanNet200 | | S3DIS |
|---|---|---|---|---|---|
| | Val | Test | Val | Test | Area5 |
| PCM (Zhang et al., 2024) | - | - | - | - | 63.4 |
| PointNeXt (Qian et al., 2022) | 71.5 | 71.2 | - | - | 70.5 |
| MinkUNet (Choy et al., 2019) | 72.2 | 73.6 | 25.0 | 25.3 | 65.4 |
| ST (Lai et al., 2022) | 74.3 | 73.7 | - | - | 72.0 |
| PTv2 (Wu et al., 2022) | 75.4 | 74.2 | 30.2 | - | 71.6 |
| OctFormer (Wang, 2023) | 75.7 | 76.6 | 32.6 | 32.6 | - |
| Point Mamba (Liu et al., 2024b) | 75.7 | - | - | - | - |
| OA-CNNs (Peng et al., 2024) | 76.1 | 75.6 | 32.3 | 33.3 | 71.1 |
| Swin3D (Yang et al., 2023) | 76.4 | - | - | - | 72.5 |
| PTv3 (Wu et al., 2024) | 77.5 | 77.9 | 35.2 | 37.8 | 73.4 |
| MEEPO(OURS) | **78.0** | **78.4** | **36.0** | **38.5** | **73.5** |

Table 4: Outdoor semantic segmentation comparison on nuScenes.

| Method | nuScenes | |
|---|---|---|
| | val | test |
| AF2S3Net (Cheng et al., 2021) | 62.2 | 78.0 |
| MinkUNet (Choy et al., 2019) | 73.3 | - |
| Cylinder3d (Zhu et al., 2021) | 76.1 | 77.2 |
| SPVNAS (Tang et al., 2020) | 77.4 | - |
| RPVNet (Xu et al., 2021) | 77.6 | - |
| RangeFormer (Kong et al., 2023) | 78.1 | 80.1 |
| SphereFormer (Lai et al., 2023) | 78.4 | 81.9 |
| OA-CNNs (Peng et al., 2024) | 78.9 | - |
| PTv2 (Wu et al., 2022) | 80.2 | 82.6 |
| PTv3 (Wu et al., 2024) | 80.4 | 82.7 |
| MEEPO (Ours) | **80.8** | **82.8** |

Table 5: Latency, parameters and accuracy comparison on ScanNet.

| Method | Lat. (ms) | Params. (M) | mIoU |
|---|---|---|---|
| Point Mamba (Liu et al., 2024b) | 280 | 109.5 | 75.7 |
| PTv2 (Wu et al., 2022) | 296 | **12.8** | 75.4 |
| PTv3 (Wu et al., 2024) | 190 | 46.2 | 77.5 |
| OctFormer (Wang, 2023) | 133 | 44.0 | 75.7 |
| OA-CNN (Peng et al., 2024) | 127 | 51.5 | 76.1 |
| MEEPO (Ours) | **110** | 45.6 | **78.0** |

Table 6: Performance on ScanNet data efficient benchmark.

| Method | Limited Reconstruction | | | | Limited Annotation | | | |
|---|---|---|---|---|---|---|---|---|
| | 1% | 5% | 10% | 20% | 20 | 50 | 100 | 200 |
| MinkUNet (Choy et al., 2019) | 26.0 | 47.8 | 56.7 | 62.9 | 41.9 | 53.9 | 62.2 | 65.5 |
| PTv2 (Wu et al., 2022) | 24.8 | 48.1 | 59.8 | 66.3 | 58.4 | 66.1 | 70.3 | 71.2 |
| PTv3 (Wu et al., 2024) | 25.8 | 48.9 | 61.0 | 67.0 | 60.1 | 67.9 | 71.4 | 72.7 |
| MEEPO (Ours) | **26.4** | **50.9** | **62.3** | **68.1** | **61.9** | **68.8** | **72.3** | **74.4** |

# 5 EXPERIMENTS

In this section, we begin by briefly describing our implementation details (Sec. 5.1). Then, we compare MEEPO with state-of-the-art (SOTA) methods (Sec. 5.2) and ablate our proposed method (Sec. 5.3). Due to space limitation, detailed implementation details, more quantitative and qualitative results and additional ablations are presented in the Appendix A.

## 5.1 IMPLEMENTATION DETAILS

**Datasets.** We evaluate our proposed method on several indoor and outdoor semantic segmentation datasets using mean Intersection over Union (mIoU) metric. For indoor scenes, we use ScanNet (Dai et al., 2017), its extended version ScanNet200 (Rozenberszki et al., 2022), and S3DIS (Armeni et al., 2016). For outdoor scenes, we employ nuScenes (Caesar et al., 2020).

**Training and Inference Details.** We follow all experimental settings of PTv3 (Wu et al., 2024) without any changes. For indoor segmentation, the number of epochs is 800, the learning rate is 0.006, and the weight decay is 0.05. For outdoor segmentation, the number of epochs is 50, the learning rate is 0.002, and the weight decay is 0.005. We train all our models using batch size of 12 and the AdamW optimizer. For efficiency evaluations, we use a single V100 with a batch size of 1.

## 5.2 MAIN RESULTS

**Indoor Semantic Segmentation.** Tab. 3 compares MEEPO with leading methods on the indoor ScanNet, ScanNet200, and S3DIS Area 5 cross-val datasets. MEEPO achieves new SOTA results with mIoU scores of **78.0**, **36.0**, and **73.5** on ScanNet val, ScanNet200 val, and S3DIS Area 5 cross-val, respectively, outperforming the second-best method, PTv3, by **+0.5**, **+0.8**, and **+0.1** points.

**Outdoor Semantic Segmentation.** Tab. 4 compares MEEPO with leading methods on the outdoor nuScenes val dataset. MEEPO achieves new SOTA result with mIoU score of **80.8** on nuScenes val, surpassing the second-best method, PTv3, by **0.4** points.

**Model Efficiency.** In Fig. 1, we compare the average latency and memory usage of our model with multiple leading methods on the ScanNet val dataset. As shown in Fig. 1(a) and Fig. 1(b), MEEPO exhibits better latency and memory consumption than many previous leading networks and achieves much higher performance. Remarkably, MEEPO not only outperforms the previous top-performing method, PTv3, but is also 42.1% faster and 5.53× more memory efficient. MEEPO

Table 7: **Ablation experiments on** on ScanNet for evaluating different design choices used for MEEPO. The entries marked in gray are the same, which specify the default settings.

(a) Effectiveness of Proposed Architecture

| case | params (M) | latency (ms) | mIoU |
|---|---|---|---|
| Pure CNN | **41.6** | **80** | 73.2 |
| Pure Transformer | 47.4 | 241 | 69.3 |
| Pure Mamba | 48.4 | 126 | 70.7 |
| MEEPO (Ours) | 45.6 | 110 | **78.0** |

(b) Effectiveness of CNN-Mamba Block

| number of blocks | memory (GB) | latency (ms) | mIoU |
|---|---|---|---|
| 22 | **4.9** | **110** | **78.0** |
| 20 | 5.0 | 112 | 77.7 |
| 12 | 5.6 | 117 | 77.5 |
| 8 | 6.1 | 122 | 77.2 |
| 4 | 10.3 | 132 | 77.3 |
| 0 | 29.5 | 153 | 77.4 |

(c) Effectiveness of Causal-Free Conv1D

| case | mIoU |
|---|---|
| Causal Conv1D | 77.5 |
| Causal-Free Conv1D | **78.0** (+0.5) |

(d) Effectiveness of Bidirectional Strided SSM

| case | mIoU |
|---|---|
| Standard SSM | 77.2 |
| Bidirectional SSM | 77.3 (+0.1) |
| Strided SSM | 77.7 (+0.5) |
| Bidirectional Strided SSM | **78.0** (+0.8) |

(e) Optimal Stride for Bidirectional Strided SSM

| stride | mIoU |
|---|---|
| 1 | 77.5 |
| 2 | **78.0** |
| 4 | 77.7 |
| 8 | 77.6 |
| 16 | 77.5 |

(f) Compatibility of Our Proposed Modules

| case | mIoU |
|---|---|
| baseline (Pure CNN network) | 73.2 |
| + CNN-Mamba blocks | 76.9 (+3.7) |
| + Causal-Free Conv1D | 77.4 (+4.2) |
| + Bidirectional Strided SSM | 78.0 (+4.8) |

particularly excels at ultra-long-range modeling, owing to its use of Mamba, which scales linearly with respect to the number of input points. As illustrated in Fig.1(d), MEEPO achieves $12\times$ reduction in FLOPs when handling scenes containing $131,072$ points. Note that Tab. 5 gives the exact numbers corresponding to data points in Fig. 1(a) and Fig. 1(b).

**Data Efficiency.** In Tab. 6, we compare MEEPO with leading methods on the ScanNet data efficiency benchmark, which evaluates models with limited reconstructions and annotations. The "Limited Reconstruction" setting uses a fraction of the available 3D reconstructions, while the "Limited Annotation" setting restricts the number of annotated points per scene. As shown, MEEPO outperforms the second-best method, PTv3, by **0.6**, **2.0**, **1.3**, and **1.1** points at 1%, 5%, 10%, and 20% reconstructions, and by **1.8**, **0.9**, **0.9**, and **1.7** points at 20, 50, 100, and 200 annotations, respectively.

## 5.3 ABLATION EXPERIMENTS

In this subsection, we conduct ablations for all components of our architecture using the ScanNet (Dai et al., 2017) val dataset. All experimental settings follow the settings used in the main results.

**Effectiveness of Our Proposed Architecture.** Aside from the qualitative comparisons presented in Fig. 2 and Fig. 3, we also quantitatively compare our proposed method with other model architectures. To ensure similar parameter counts, the channel sizes of these networks are adjusted accordingly. As shown in Tab. 7(a), MEEPO performs much than single-operator networks.

**Effectiveness of CNN-Mamba Block.** In Tab. 7(b), we progressively replace some of the CNN-Mamba blocks in MEEPO with CNN-Transformer blocks. The results show that increasing the number of CNN-Transformer blocks always leads to higher latency, more memory usage and reduced performance. This confirms the effectiveness of our proposed CNN-Mamba block, which can efficiently and effectively integrate local and contextual modeling for point cloud segmentation.

**Effectiveness of Causal-Free Mamba.** The original Mamba uses causal depthwise convolution to preprocess input tokens before passing them to the SSM module. While this makes sense for sequence modeling, its necessity for 3D vision tasks, which lack causality, is unclear. To investigate this, we experiment with replacing the causal convolution with normal convolution. As shown in Tab. 7(c), causal convolution results in a performance improvement of **+0.5** mIoU.

**Effectiveness of Bidirectional Strided SSM.** In Tab. 7(d), we demonstrate the effectiveness of our proposed *Bidirectional Strided SSM*. As shown, it outperforms both the standard SSM and its bidirectional variant, resulting in a performance improvement of **+0.8** mIoU. Additionally, we ablate the optimal stride for this module in Tab. 7(e), which shows that a stride of 2 yields the best performance.

**Compatibility of Our Proposed Modules.** In addition to ablating our proposed modules individually, we perform an additive ablation study in Tab. 7(f) to demonstrate the compatibility of the enhancements. Incorporating CNN-Mamba blocks results in a significant improvement of **+3.7** mIoU. Further upgrades to the standard Mamba module, namely introducing causal-free convolutions and employing multi-directional scanning, provide additional gains of **+0.5** and **+0.6** mIoU, respectively.

## 6 RELATED WORK

**Architectures for point cloud segmentation** fall into three main categories: point-based (Thomas et al., 2019; Qi et al., 2017a;b; Ma et al., 2022), voxel-based (Maturana & Scherer, 2015; Song et al., 2017), and projection-based (Chen et al., 2017; Lang et al., 2019; Li et al., 2016; Su et al., 2015). Although they differ in pre-processing strategies, all these methods are designed with careful consideration of the unique characteristics of point clouds. They mainly differ in how they integrate local and contextual features and manage irregular point distributions. Recently, transformer-based models (Wu et al., 2024; Robert et al., 2023; Yang et al., 2023; Lai et al., 2022) have emerged in this field, achieving higher accuracy but suffering from significant time and memory complexities relative to the size of the point cloud. To address this, more efficient attention mechanisms, such as vector attention (Zhao et al., 2020), grouped vector attention (Wu et al., 2022), local window-based attention (Lai et al., 2022), and memory-efficient attention (Yang et al., 2023), have been developed. However, these strategies approximate original attention, resulting in a loss of global modeling capability (Shen et al., 2021), which may impede performance on long range modeling task like point cloud segmentation. Our research examines the strengths and weaknesses of several widely-used architectures for point cloud segmentation and introduces a novel architecture that effectively integrates their best features, culminating in a new state-of-the-art model for point cloud segmentation.

**State space models (SSMs)** (Gu & Dao, 2024; Wang et al., 2024b; Lieber et al., 2024) have emerged as a promising alternative to Transformers (Vaswani et al., 2017) in natural language processing (NLP) for capturing long-range dependencies. Unlike Transformers that scale quadratically with sequence length, SSMs (Gu et al., 2022; Nguyen et al., 2022; Smith et al., 2023) achieve linear scaling during inference. The seminal Mamba model (Gu & Dao, 2024) greatly improves the performance and efficiency of SSM by introducing input-specific parameterization and a scalable, hardware-optimized method, allowing it to outperform Transformers for the first time. Driven by the success of SSMs in NLP, recent works have also explored their application to visual tasks. S4ND (Nguyen et al., 2022) marks the introduction of SSM modules for processing visual data across 1D, 2D, and 3D domains. Subsequent works, such as VMamba (Liu et al., 2024c), Vim (Zhu et al., 2024), and Bi-Mamba+ (Liang et al., 2024a), address the directional sensitivity in SSMs with bi-directional and cross-scan mechanisms, allowing SSMs to achieve performance that rivals that of CNN and Transformer models. Mamba-based models have also delivered impressive performance in many other vision tasks, such as image segmentation (Xing et al., 2024; Liu et al., 2024a), image synthesis (Yan et al., 2024), graph modeling (Wang et al., 2024a), and low-level vision (Guo et al., 2024). Despite some initial attempts (Liu et al., 2024b) to apply Mamba for point cloud segmentation, these models have all failed to outperform existing architectures. In this work, we identify the shortcomings of SSM when applied to this task and propose simple solutions to greatly improve its performance.

## 7 CONCLUSION

In this work, we present a detailed analysis of existing network architectures for point cloud segmentation, highlighting their strengths and weaknesses. This evaluation provides valuable insights into designing more efficient and effective architectures for this task. Our findings emphasize the crucial role of *spatial locality* and *robust contextual understanding* in achieving strong performance. Specifically, we identify convolution and the Mamba module as essential components for efficient and accurate point cloud segmentation. Convolution provides spatial locality, while the Mamba module enhances the understanding of context. Additionally, we improve the standard Mamba module by removing the causality constraint and introducing *Bidirectional Strided SSM*, which further enhances its ability to capture and utilize contextual information. Following these design principles and applying targeted optimizations, we introduce MEEPO, a novel architecture that outperforms previous state-of-the-art models across multiple key benchmark datasets and efficiency metrics.

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

# A APPENDIX

For a thorough understanding of our proposed MEEPO, we have compiled a detailed Appendix. The table of contents below offers a quick overview and will guide to specific sections of interest.

CONTENTS

## A.1 IMPLEMENTATION DETAILS

We implement our method using the Pointcept (Contributors, 2023) codebase. Detailed specifications of our implementation are provided in this section.

Table 8: Indoor sem. seg. settings.

| Config | Value |
|---|---|
| optimizer | AdamW |
| scheduler | Cosine |
| criteria | CrossEntropy (1) |
| | Lovasz (1) |
| learning rate | 6e-3 |
| block lr scaler | 0.1 |
| weight decay | 5e-2 |
| batch size | 12 |
| datasets | ScanNet / S3DIS |
| warmup epochs | 40 |
| epochs | 800 |

Table 9: Outdoor sem. seg. settings.

| Config | Value |
|---|---|
| optimizer | AdamW |
| scheduler | Cosine |
| criteria | CrossEntropy (1) |
| | Lovasz (1) |
| learning rate | 2e-3 |
| block lr scaler | 0.1 |
| weight decay | 5e-3 |
| batch size | 12 |
| datasets | nuScenes |
| warmup epochs | 2 |
| epochs | 50 |

**Training Settings.** The experimental settings for indoor and outdoor semantic segmentation are outlined in Tab. 8 and Tab. 9. The numbers in brackets indicate the relative weight assigned to each criterion in the loss. The main differences between indoor and outdoor settings are in the learning rate, weight decay, warmup epochs and training epochs used.

Table 10: Model settings.

| Config | Value |
|---|---|
| positional encoding | None |
| embedding depth | 2 |
| embedding channels | 32 |
| no. of layers in Local Perceiver | 1 |
| no. of layers in Channel Modulator | 2 |
| encoder depth | [2, 2, 6, 2] |
| encoder channels | [64, 128, 256, 512] |
| encoder num heads | [4, 8, 16, 32] |
| decoder depth | [1, 1, 1, 1] |
| decoder channels | [64, 64, 128, 256] |
| decoder num heads | [4, 4, 8, 16] |
| down stride | [$\times 2$, $\times 2$, $\times 2$, $\times 2$] |
| drop path | 0.3 |

**Model Settings.** Detailed model configurations of our MEEPO are listed in Tab. 10.

Table 11: Data augmentations.

| Augmentations | Parameters | Indoor | Outdoor |
|---|---|:---:|:---:|
| random dropout | dropout ratio: 0.2, p: 0.2 | ✓ | - |
| random rotate | axis: z, angle: [-1, 1], p: 0.5 | ✓ | ✓ |
|  | axis: x, angle: [-1 / 64, 1 / 64], p: 0.5 | ✓ | - |
|  | axis: y, angle: [-1 / 64, 1 / 64], p: 0.5 | ✓ | - |
| random scale | scale: [0.9, 1.1] | ✓ | ✓ |
| random flip | p: 0.5 | ✓ | ✓ |
| random jitter | sigma: 0.005, clip: 0.02 | ✓ | ✓ |
| elastic distort | params: [[0.2, 0.4], [0.8, 1.6]] | ✓ | - |
| auto contrast | p: 0.2 | ✓ | - |
| color jitter | std: 0.05; p: 0.95 | ✓ | - |
| grid sampling | grid size: 0.02 (indoor), 0.05 (outdoor) | ✓ | ✓ |
| sphere crop | max points: 102400 | ✓ | - |
| normalize color | p: 1 | ✓ | - |

**Data Augmentation.** Data augmentations used for training MEEPO are detailed in Tab. 11.

A.2    ADDITIONAL QUANTITATIVE RESULTS

Table 12: Results on ScanNet200 for semantic segmentation.

| Method | Val | | | |
|---|:---:|:---:|:---:|:---:|
|  | Head | Comm. | Tail | All |
| MinkowskiNet (Choy et al., 2019) | 48.3 | 19.1 | 7.9 | 25.1 |
| SparseUNet (Wu et al., 2023) | - | - | - | 28.8 |
| LGround (Rozenberszki et al., 2022) | 51.5 | 22.7 | 12.5 | 28.9 |
| PTv2 (Wu et al., 2022) | - | - | - | 29.3 |
| OA-CNNs (Peng et al., 2024) | 51.3 | 28.0 | 17.7 | 32.3 |
| PTv3 (Wu et al., 2024) | 56.5 | 30.1 | 19.3 | 35.2 |
| MEEPO | **56.6** | **30.7** | **20.7** | **36.0** |

**Class Imbalance Analysis on ScanNet200 (Rozenberszki et al., 2022).** In Tab. 12, we compare MEEPO with other leading methods on the *head*, *common*, and *tail* subsets of the ScanNet200 validation benchmark. This comparison offers a more nuanced understanding of performance across different levels of class imbalance. As shown, MEEPO achieves improvements of +0.2, +0.6, and +1.4 mIoU on these three subsets, respectively. Notably, it performs much better on the most challenging *tail* subset, indicating its potential in handling long-tail distributions.

Table 13: Additional Evidences of Mamba's Importance.

| Case | Params (M)↓ | mIoU↑ |
|---|:---:|:---:|
| MEEPO | 45.6 | 78.0 |
| Replace Mamba with Attention | 42.7 | 77.4 |
| Remove Mamba | 38.2 | 73.5 |
| Use one additional sparse conv. layer | 69.0 | 75.7 |
| Use one additional block in each stage | 52.7 | 74.7 |
| Increase input channel size from 32 to 36 | 48.3 | 74.2 |

**Additional Evidences of Mamba's Importance.** To emphasize Mamba's importance, we also test alternative methods to scale up the models without it. The results in Tab. 13 show that none of the MEEPO variants without Mamba outperform the original configuration, clearly demonstrating its critical role in providing contextual modeling for point cloud segmentation.

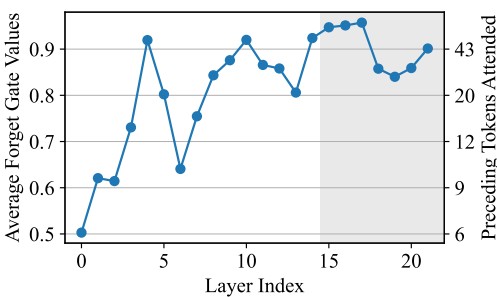

Figure 7: The average of forget gate values in different layers of MEEPO.

**Additional Evidence of Mamba's Local Bias.** Fig. 7 shows average forget gate values for each layer and their attenuation effects. Let $\bar{\mathbf{A}}_{layer}$ be the average forget gate value of a layer. The number of preceding tokens attended is computed by taking $\max_k \bar{\mathbf{A}}_{layer}^k < 0.01$, which corresponds to the number of tokens that accounts for more than 1% of the current value. In shallow layers, $\bar{\mathbf{A}}_{layer}$ tend to be small, indicating that each token primarily focuses on the preceding tokens in its vicinity, thus demonstrating strong local bias. In deeper layers, the average ranges from approximately 0.8 to 1.0, suggesting a broader receptive field for each token. These findings confirm our analysis that while having broad learnable adaptive receptive fields, Mamba tends to be locally biased (Han et al., 2024).

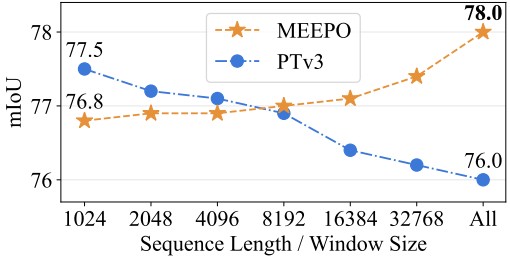

Figure 8: Performance comparison between MEEPO and PTv3 in Processing Long Context.

**Effectiveness of Mamba in Processing Long Context.** Due to their linear complexity, Mamba-based networks can efficiently process entire point clouds without requiring window partitioning. Similarly, the proposed MEEPO architecture operates on complete point clouds without the need for windowing. As shown in Fig. 1(a) and (b), despite using all points, MEEPO is still 42.1% faster and 5.53 times more memory-efficient than the best performance of PTv3, which uses a sequence length of 1024. Nonetheless, We investigate in Fig. 8 whether such window splitting can have performance benefit. As shown in the figure, using more points progressively improves results, with optimal performance achieved when using all points. This indicates that Mamba is highly effective in processing long context. This is a highly beneficial property as processing all points offers a complete view of the scene, avoiding complex approximations and ensuring that no details are overlooked. Conversely, segmenting a point cloud into windows may conceal crucial interactions across boundaries, resulting in a potentially incomplete or inaccurate scene representation.

## A.3 ADDITIONAL QUALITATIVE RESULTS

| Input | Ground Truth | PTv3's predictions | Meepo's predictions |

Legend:
wall · floor · cabinet · bed · chair · sofa · table · door · window · bookshelf
picture · counter · desk · curtain · fridge · shower curtain · toilet · sink · bathtub · other furniture

Figure 9: Comparison between MEEPO's and PTv3's (Wu et al., 2024) predictions. Black color are unlabelled points. Red boxes with dash-dotted lines are wrong predictions by PTv3.

A.4    LIMITATIONS

While our work has significantly advanced the performance of point cloud segmentation models, many challenges and opportunities for improvement remain. For instance, although MEEPO exhibits notable efficiency gains, there is still room for further optimization to enhance its computational and memory efficiency. Additionally, segmentation quality, especially in handling fine details and complex geometries within point clouds, can be further improved. Enhancing the model's ability to accurately segment objects in diverse and densely populated scenes is another critical area for future research. Moreover, the potential of pretraining through self-supervised learning is unexplored in this work. Leveraging large-scale unlabeled point cloud data for pretraining could help the model learn more robust and generalizable features, ultimately boosting performance across various tasks and datasets. Future work should explore and integrate self-supervised learning techniques to harness this potential fully. Addressing these challenges will ensure that Mamba-based architectures continue to evolve and set new benchmarks in the field of point cloud segmentation.

A.5    BROADER IMPACTS

**Accessibility and Resource Efficiency.** By demonstrating that our proposed method, MEEPO can achieve superior performance with reduced latency and memory consumption, this research promotes the development of more accessible and resource-efficient machine learning models. This is particularly important for applications in resource-constrained environments, such as mobile and embedded systems, where computational power and memory are limited. As a result, more organizations and developers can leverage advanced point cloud segmentation techniques without requiring extensive computational resources.

**Environmental Impact.** The reduction in computational cost and memory usage directly translates to lower energy consumption. Given the growing concern over the environmental impact of large-scale machine learning models, MEEPO's efficiency can contribute to more sustainable AI practices. By minimizing the energy required for training and inference, this work aligns with global efforts to reduce the carbon footprint of technology.

A.6    COMPUTE RESOURCES

We run all experiments on a cluster with a mix of RTX3090, RTX4090 or 32GB V100 and 40GB A100 GPUs.

A.7    REPRODUCIBILITY

Our main results can be fully reproduced by running the training and evaluation scripts given in the attached code.

