# OpenReview forum: "Exploring contextual modeling with linear complexity for point cloud segmentation"
_ICLR.cc/2025/Conference — Submitted to ICLR 2025_

### Official Review · Reviewer_fEVg · 2024-10-17

**Soundness:** 2
**Presentation:** 3
**Contribution:** 2
**Rating:** 6
**Confidence:** 4

**Summary:**

Point cloud segmentation is crucial for 3D understanding, traditionally tackled using CNNs or Transformers. Recently, Mamba has emerged as an efficient solution for contextual modeling, though it has struggled to outperform leading CNN and Transformer methods. This work identifies key requirements for effective segmentation: strong spatial locality and robust contextual understanding. Enhancing Mamba, the authors remove causality and introduce a Strided Bidirectional SSM to address directional biases in unordered point clouds. The resulting architecture, MEEPO, merges the strengths of CNNs and Mamba, achieving up to +0.8 mIoU over state-of-the-art methods on benchmarks like ScanNet and nuScenes, with notable gains in speed and memory efficiency.

**Strengths:**

1. The paper is well-structured with a clear analysis-driven approach.
2. The experimental results are impressive.

**Weaknesses:**

1. In lines 238-241, it's unclear why Mamba is considered effective for local processing, given its design goal of modeling long sequences with linear complexity. The reference to a "locally-biased forget gate" needs further explanation and detailed analysis or statistical data beyond visualization.

2. In lines 300-302, the statement that long-range attention is unnecessary may only partially reflect the issue. The core reason might be the sparsity of 3D point clouds. Transformers generally require extensive data and prolonged training to surpass CNNs, so it might be more accurate to say that insufficient data limits the full potential of Transformers rather than implying long-range attention is irrelevant.

3. The innovations appear somewhat limited, with minimal structural changes. The modifications, such as Causal-Free Mamba and Bidirectional Strided SSM, feel more like tricks.

4. Presenting results on downstream perception tasks like segmentation and detection would add further value.

Minor Errors:
- Line 251: Fig.4 should be Fig.5.
- clarify the point cloud processing order in Mamba. Is it similar to PTv3's order?

**Questions:**

see above weakness

---

> ### Author Response · Authors · 2024-11-13
>
> Dear Reviewer,
>
> Thanks for the comment. We would like to clarify that this paper is an improved version of a rejected paper previously submitted to *NeurIPS 2024*. However, we absolutely did NOT submit to *AAAI 2025*. Therefore, we can think of two possibilities:
>
> (1) The *NeurIPS* paper has been mistakenly identified as an *AAAI* submission.
>
> (2) Someone else has resubmitted our *NeurIPS* paper to *AAAI* without our knowledge.
>
> Could you please kindly check and confirm that you have indeed seen this paper in *AAAI2025*?  If so, this is a very serious academic issue and we will need to involve the help of PC to help investigate this issue.

---

> > ### Comment · Reviewer_fEVg · 2024-11-13
> > **Sincerely sorry for the mistake, i'll update my comments soon**
> >
> > I'm sorry that I have mistaken the NeurIPS paper as an AAAI submission. I will update my comments soon.

---

> ### Author Response · Authors · 2024-11-20
>
> Thank you for your detailed review. We are glad that you find our paper to be well-structured and results to be impressive! Below are our response to the weaknesses:
>
> **[Q1] More explanation for Mamba's local bias**
> > The effectiveness of Mamba for local processing due to locally-biased forget gate has been rigorously examined in the recent NeurIPS 2024 paper "Demystify Mamba in Vision: A Linear Attention Perspective" by Han et al. In Sec 4.2 and Fig. 4, the authors demonstrate that Mamba tends to focus predominantly on recent tokens. To give more statistical data to support this claim, we reproduce their findings and include a similar plot for point cloud segmentation in Fig. 7 of our updated paper's appendix (line 822). Additionally, we also add this citation to our updated paper.
>
> **[Q2] Improved explanation for performance degradation of PTv3 with respect to increasing window size**
> > Thanks for the suggestion! We agree with your intuition and have improved the corresponding sections on lines 234 and 298 of the updated paper to emphasize this point.
>
> **[Q3]: Regarding minimal structural change to Mamba**
> > We would like to clarify that our paper extends beyond providing structural changes to Mamba. It offers a comprehensive analysis of architectural properties of commonly used point cloud segmentation operators, investigates the reasons behind Mamba-based networks' underperformance in this task, and proposes simple yet effective solutions to address these issues. Our research provides valuable insights for designing efficient and effective networks, and our experiments demonstrate that these solutions significantly mitigate the identified problems. While more complex approaches are possible and could be explored in future work, we favor simple solutions for their clarity, ease of explanation, and broader applicability.
>
> **[Q4] Additional 3D object detection experiment**
> > Following your suggestion, we provide an additional 3D object detection result on ScanNet v2 dataset:
> | backbone | mAP@0.25 |
> | -------- | ------- |
> | PTv3  | 71.3   |
> | MEEPO | 72.2 (+0.9) |
>
> **Figure reference fix**
> > Thanks for the suggestion! We have corrected the figure reference error in the updated paper.
>
> **Clarification of Mamba's point cloud processing order**
> > The order is the same as the one used in PTv3. We have added this clarification to line 189 of the updated paper.

---

> ### Comment · Reviewer_fEVg · 2024-11-25
>
> The authors' responses have addressed my concerns well, i'll slightly improve my score

---

> > ### Author Response · Authors · 2024-11-25
> >
> > Thank you for your response! We’re delighted to hear that our reply addressed your concerns and appreciate your recognition of our work.

---

### Official Review · Reviewer_XaY8 · 2024-10-26

**Soundness:** 3
**Presentation:** 3
**Contribution:** 3
**Rating:** 6
**Confidence:** 4

**Summary:**

This paper analyzes several popular neural network architectures, including CNN-based and Transformer-based designs, as well as the recently introduced Mamba model. It proposes a new point cloud segmentation architecture, MEEPO, which combines the strengths of CNNs and Mamba, surpassing previous state-of-the-art methods like PTv3 on multiple key benchmark datasets.

**Strengths:**

1. The model is highly efficient and achieves accuracy that exceeds prior state-of-the-art methods.
2. The design of the Strided Bidirectional SSM effectively enhances the model's understanding of spatial relationships.

**Weaknesses:**

1. The proposed hyper-structure is based on the analysis of existing architectures, which makes the technical innovation somewhat limited.

**Questions:**

I am unsure whether the authors of this paper are aware of *Point Cobra*. I would like the authors to address the numerous similarities between the two papers and provide an explanation.

1. While the new model structure aligns with the previous version (NIPS2024), there are differences in latency and memory usage. I am interested in understanding the source of these improvements.

2. This paper introduces an additional CNN module to capture the local features of the point cloud. Could the Causal-Free Conv1D in the Mamba block be removed, given that these two modules appear to serve the same role in the model?

**Details Of Ethics Concerns:**

Sorry for misidentifying the NeurIPS paper as an AAAI submission. The author’s response has alleviated my concerns about duplicate submission or plagiarism.

---

> ### Author Response · Authors · 2024-11-13
>
> Dear Reviewer,
>
> Thanks for the comment. We would like to clarify that this paper is an improved version of a rejected paper previously submitted to *NeurIPS 2024*. However, we absolutely did NOT submit to *AAAI 2025*. Therefore, we can think of two possibilities:
>
> (1) The *NeurIPS* paper has been mistakenly identified as an *AAAI* submission.
>
> (2) Someone else has resubmitted our *NeurIPS* paper to *AAAI* without our knowledge.
>
> Could you please kindly check and confirm that you have indeed seen this paper in *AAAI2025*?  If so, this is a very serious academic issue and we will need to involve the help of PC to help investigate this issue.

---

> > ### Comment · Reviewer_XaY8 · 2024-11-13
> > **Point Cobra is a paper reviewed in Nips 2024**
> >
> > Sorry for misidentifying the NeurIPS paper as an AAAI submission. The author’s response has alleviated my concerns about duplicate submission or plagiarism.
> >
> > I still have some questions regarding the method:
> > 1. While the new model structure aligns with the previous version (NIPS2024), there are differences in latency and memory usage. I am interested in understanding the source of these improvements.
> > 2. This paper introduces an additional CNN module to capture the local features of the point cloud. Could the Causal-Free Conv1D in the Mamba block be removed, given that these two modules appear to serve the same role in the model?

---

> > > ### Author Response · Authors · 2024-11-20
> > >
> > > **[Q1] Differences from NeurIPS version**
> > > > They are not exactly the same. Since then, we have made several hyper-parameter optimizations, such as reducing the MLP ratio and adding an extra layer to the stem.
> > >
> > > **[Q2] Can causal conv in SSM be removed?**
> > > > Following your suggestion, we try removing the convolution. However, it gives a worse mIoU score on ScanNet (77.3 vs 77.5)

---

### Official Review · Reviewer_wi14 · 2024-11-03

**Soundness:** 3
**Presentation:** 3
**Contribution:** 3
**Rating:** 6
**Confidence:** 4

**Summary:**

This paper presents a novel architecture named MEEPO, designed for point cloud segmentation with a focus on efficient contextual modeling. It introduces Mamba, a state-space model (SSM) that achieves linear complexity compared to traditional Transformers, which have quadratic complexity. The authors identify Mamba's limitations, such as enforced causality and directional bias, and propose solutions through causal-free convolutions and bidirectional strided state-space modeling. MEEPO outperforms previous models, like PTv3, in accuracy, latency, and memory efficiency across several benchmark datasets (e.g., ScanNet, S3DIS, nuScenes).

**Strengths:**

1.	MEEPO introduces a new method by integrating CNN and Mamba components, achieving good contextual modeling with reduced computational costs.
2.	The paper thoroughly explores the limitations of existing architectures (CNN, Transformer, and SSM) and provides comprehensive ablations and comparisons to validate its design choices.
3.	The introduction of causal-free convolutions and bidirectional strided SSM in Mamba addresses some original limitations in Mamba for this task,.

**Weaknesses:**

1.	In Section 4.2, the authors claim that they propose a 'Bidirectional Strided SSM' method, however, Bidirectional SSMs have already been proposed in works like [1][2] to address the issue of forgetting unidirectional sequences. I believe that not referencing these articles while claiming authorship is not rigorous. Additionally, what baffles me is that in Table 7, a comparison is made for strides ranging from 2 to 16, showing a decreasing trend in performance. So, why not similarly compare and discuss the results for a stride of 1?
2.	The manuscript presents in Fig. 4 the mIoU performance of the Transformer-based PTv3 under different range windows. The advantage of the Mamba model lies in its better memory capabilities for long sequences. The authors emphasize this advantage in line 240, but they do not provide specific data to demonstrate the validation of this conclusion in point cloud data  (the author can show a similar format figure as Fig.4).
3.	Table 7 in the article indicates that the parameter volume of CNN+Mamba is smaller than that of pure Mamba. However, based on the author's description, compared to pure Mamba, the author has added a CNN module to Mamba and employed a bidirectional mechanism for computation. Why the parameter count is lower than pure Mamba needs further clarification from the author.
4.	In line 251, the referred figure is not Fig.4, it should be Fig. 5.


[1] Bi-Mamba+: Bidirectional Mamba for Time Series Forecasting
[2] Vision Mamba: Efficient Visual Representation Learning with Bidirectional State Space Model

**Questions:**

please refer to the weakness

---

> ### Author Response · Authors · 2024-11-20
>
> Thank you for your detailed review. We are glad that you find our analysis of existing architectures to be thorough and comprehensive! Below are our response to the weaknesses:
>
> **[W1] Missing Citation**
> > We have already cited Vision Mamba in our paper. We apologize for omitting Bi-Mamba+ and have now added this citation to the related work section (line 518) in the updated paper.
>
> **[W1] Additional comparison with stride 1**
> > Using a stride of 1 is equivalent to repeating the Bidirectional SSM twice. Following your suggestion, we include this setting to Tab. 7(e) of the updated paper, which confirms that strided computation matters.
>
> **[W2] Data demonstrating Mamba's advantage in performance and efficiency**
> > Mamba's performance and efficiency advantages are already illustrated in Fig. 1(a) and (b), which show that our proposed Mamba model, MEEPO, outperforms PTv3 at its best performance with a sequence length of 1024 (78.0 vs. 77.5), while being 42.1% faster and 5.53x more memory-efficient.
> >
> > &nbsp;
> >
> > In contrast, Fig. 4 primarily focuses on PTv3's performance degradation with increasing sequence length. Comparable data for Mamba is not presented because window partitioning is unnecessary for our model due to its linear complexity. In fact, our proposed MEEPO utilizes all points. Nevertheless, to provide a similar comparative analysis, we implement a modified version of MEEPO with window partitioning. The results, included in Fig.8 of the updated paper's appendix (line 841), demonstrate that MEEPO can effectively leverage long contexts, showing progressive performance improvements with increased point utilization.
>
> **[W3] Clarification on parameter count in Tab. 7**
> > The difference in parameter count is due to different channel sizes, which are adjusted to make them comparable. When using the same channel size, these different operators vary significantly in their parameter counts. For example, a 3D convolution operator has many more parameters than attention or SSM operators when using the same channel size. We have updated lines 196 and 469 to clarify this methodology.
>
> **[W4] Figure reference fix:**
> > Thanks for the suggestion! We have fixed this error in the updated paper

---

### Official Review · Reviewer_HJ9v · 2024-11-03

**Soundness:** 3
**Presentation:** 3
**Contribution:** 3
**Rating:** 5
**Confidence:** 4

**Summary:**

This paper focuses on the 3D point cloud segmentation task. It first compares the performance of three types of blocks: CNN, Transformer, and Mamba, and then improves the existing Mamba architecture. The authors provide extensive visualization results and ablation experiments.

**Strengths:**

[S1] It is meaningful to compare the performance of the three types of blocks—CNN, Transformer, and Mamba—on the same task, point cloud segmentation.
[S2] The paper is detailed and easy to read.
[S3] The paper conducts detailed ablation experiments on the newly proposed module.
[S4] The paper achieves state-of-the-art results on multiple datasets.

**Weaknesses:**

[W1] The comparison of CNN, Transformer, and Mamba blocks lacks strong evidence. I believe that concluding "CNN is more effective for local modeling and Transformer is better at handling contextual information" based on just one example for each is insufficient.
[W2] The proposed method is more like a small fix to Mamba-based method.
[W3] The two solutions proposed in this paper: 1) the casual-free block is described in very little detail and lacks a clear explanation; 2) bidirectional SSM seems to have already been introduced in VisionMamba (Zhu 2024), and this paper's work only adds n-stride.

**Questions:**

[Q1] Could the authors provide more detailed explanations of W1, such as whether there is theoretical evidence to support it or if there are sufficiently extensive experiments across multiple datasets?
[Q2] Is the Bidirectional SSM mentioned by the authors consistent with VisionMamba (Zhu 2024)? I do not see a bidirectional structure in Fig(6b), only the Strided SSM.
[Q2.5] If the answer of Q2 is yes, the result you reported in the second line of Tab(7d) shows that this structure can only bring 0.1% increasement, which seems inconsistent with the effects reported in VisionMamba (Zhu 2024) regarding this structure. From an efficiency perspective, is the introduction of the indirection necessary if it only brings a 0.1% improvement?
[Q3] The casual-free conv block is not clearly explained. If this is something new you are proposing, it would be helpful to include a more detailed explanation or illustration.
[Q4] Standardize the notation for SSM. In the abstract and introduction, it is referred to as "Strided Bidirectional SSM," while later it is called "Bidirectional Strided SSM."
[Q5] There’s something wrong of your hyperlink in 3.3. It should be Fig 5 but yours is Fig 4.

---

> ### Author Response · Authors · 2024-11-20
>
> Thank you for your detailed review. We are glad that you find our analysis to be meaningful! Below are our response to the weaknesses and questions:
>
> **[W2] Regarding simple fix to Mamba**
> > We would like to clarify that our paper extends beyond providing simple fixes to Mamba. It offers a comprehensive analysis of architectural properties of commonly used point cloud segmentation operators, investigates the reasons behind Mamba-based networks' underperformance in this task, and proposes simple yet effective solutions to address these issues. Our research provides valuable insights for designing efficient and effective networks, and our experiments demonstrate that these solutions significantly mitigate the identified problems. While more complex approaches are possible and could be explored in future work, we favor simple solutions for their clarity, ease of explanation, and broader applicability.
>
> **[Q1] Additional evidence/explanation for the respective effectiveness of CNN and Transformer for local modeling and contextual modeling**
> > Following your suggestion, we provide further statistical evidence by computing the overall improvements for the 'door' and 'table' classes, which are used to illustrate the benefits of contextual and local modeling in the visualizations. The 'door' class demonstrates the importance of contextual modeling, as door positions are heavily influenced by surrounding objects in a scene. In contrast, the 'table' class highlights the value of local modeling due to its varied shapes and structural features. The results align with our analysis and are presented below:
> | case    | 'Door' mIoU (contextual modeling) | 'Table' mIoU (local modeling) |
> | -------- | ------- | ------- |
> | Pure Transformer  | 61.1 | 51.1 |
> | Pure Mamba | 61.0 | 54.2 |
> | Pure CNN | 55.9 | 62.7 |
> >
> > &nbsp;
> >
> > Aside from that, we would like to clarify that our analysis is primarily inspired by multiple studies in 2D domains [1,2,3], which highlight the presence of architectural biases. The visualizations in Fig. 3 and Fig. 5 help to validate these intuitions and complement the quantitative experiments presented in Tab. 2. These experiments demonstrate the significance of both local and contextual modeling, as removing either component results in considerable performance degradation.
> 1. Pan et al., On the Integration of Self-Attention and Convolution, CVPR 2022
> 2. Pan et al., 3D Object Detection with Pointformer, CVPR 2021
> 3. Li et al., UniFormer: Unifying Convolution and Self-attention for Visual Recognition, TPAMI 2023
>
> **[Q2] Clarification of similarity of Bidirectional SSM with Vision Mamba**
> > Yes, the Bidirectional SSM is indeed identical to that of Vision Mamba. We apologize for the unclear Fig. 6(b) and have improved it in the updated paper.
>
> **[Q2.5] Explanation for the inconsistency with Vision Mamba**
> > We would like to clarify that there is *no* inconsistency here, as we are addressing fundamentally different tasks. While Bidirectional SSM can significantly improve performance in 2D tasks (as shown in Vision Mamba), its modest 0.1% improvement in point cloud segmentation can be attributed to the added complexity of the third dimension. In 3D space, objects exhibit much greater variability in shape and structure, often requiring more than just bidirectional scanning to fully capture all required details, particularly in dense prediction tasks like segmentation.
> >
> > &nbsp;
> >
> > Our strided scan approach offers an effective solution to address this limitation by helping to capture additional contextual information. We try using Strided SSM alone (without bidirectional scanning) but the experiment shows that  it alone *cannot* achieve the  performance achieved by the combined Bidirectional Strided SSM (77.7 vs 78.0). This result highlights that the two techniques are complementary and should be used together to achieve optimal performance. We have included the additional Strided SSM result in Table 7(d) of the updated paper.
>
> **[Q3] Explanation for Causal-Free Mamba**
> > As indicated in line 357, Causal-Free Mamba simply replaces the causal convolution with standard convolution (using torch.nn.Conv1d).
>
> **[Q4,Q5] Hyperlink and notation fix**
> >Thanks for the suggestions! We have fixed the incorrect link and notation in the updated paper.

---

> ### Author Response · Authors · 2024-11-29
> **Kind Reminder**
>
> Dear Reviewer HJ9v,
>
> We sincerely appreciate the opportunity to address your concerns. We hope our responses have adequately resolved the points you raised. As the deadline is approaching, we kindly request your feedback at your earliest convenience. If you need further clarification on any aspect of our work, please don’t hesitate to let us know.
>
> Best regards,
>
> The Authors

---

### Official Review · Reviewer_U9M8 · 2024-11-04

**Soundness:** 3
**Presentation:** 4
**Contribution:** 3
**Rating:** 8
**Confidence:** 5

**Summary:**

This paper focuses on a simple yet practical target in 3D understanding: how to enhance the accuracy of a Mamba-based framework while preserving its efficiency rooted in linear complexity. The paper seeks to achieve this through an in-depth analysis of the components that contribute to the strong performance of PTv3, summarized as contextual understanding and spatial locality. These findings lead to the two major design elements of this work: Causal-free Mamba and Bidirectional Strided SSM. Overall, it is encouraging to see the method achieve solid performance on major scene-level point cloud semantic segmentation benchmarks.

**Strengths:**

1. **[Analysis-driven Methodology]** First of all, I appreciate the paper's analysis-driven approach: rather than being experiment-driven, it presents insights based on previous work and derives its methodology from this understanding. This makes the paper easier to follow and the proposed method more convincing.

2. **[Strong performance]** The paper achieves strong performance on several major point cloud semantic segmentation benchmarks and appears to be efficient as well.

**Weaknesses:**

The insightful analysis, convincing approach, and solid performance demonstrate that this manuscript makes **a valuable empirical contribution**. However, the following weaknesses **limit its broader impact**:

1. **[Scaling up]**
**(a)** Many observations and analyses (such as the ablation on PTv3 window size) are based on training from scratch on ScanNet (1,500 samples), which is relatively small in scale. However, many model properties may change significantly when scaled up with more data. For instance, using the PTv3 window size ablation as an example, attention mechanisms are inherently adaptive to kernel size, meaning accuracy should not be negatively impacted by the window size. The degradation observed when increasing the window size beyond 1024 is likely due to insufficient data to support this adaptive capacity.
**(b)** It is good to see the Mamba framework achieve both higher accuracy and efficiency compared to previous SOTA. However, the true value of superior scratch accuracy and efficiency lies in its capacity for further scaling. I am particularly curious about the model's accuracy and efficiency when scaling up training through multi-dataset joint training, as well as its performance when scaling up parameters with larger data volumes.

2. **[Go beyond semantic segmentation]**
Why do point cloud perception and representation learning prioritize semantic segmentation? This is because it is the simplest way to evaluate the quality of learned representations using a single linear layer. However, this does not mean that research on point cloud backbones should be limited to semantic segmentation alone. The claims of the paper would be stronger if more downstream tasks, such as instance segmentation and object detection, were included. It may not be necessary to achieve the highest performance; instead, demonstrating more properties of the proposed method could be more informative.

**Questions:**

1. In Figure 1(b), PTv3 consumes even more memory compared to PTv2, which seems unusual since window-based attention should be significantly more memory-efficient than neighborhood attention. The only reason I can imagine is that FlashAttention may be disabled while using a large kernel, resulting in larger matrix multiplications. Could you provide a more detailed explanation of this setup?

2. Maybe a shorter, more impactful title could make it easier for readers to remember? The current version is too lengthy.

3. Maybe the table arrangement could be improved? I don’t recommend using resizebox, as it can lead to uneven text sizes. You might consider referring to the LaTeX source code from the PTv3 paper for tips on adjusting table formatting.

4. It might be better to reduce the use of bold text? For instance, consider changing `\textbf{Proposed Solution:}` to` \textit{Proposed solution:}`, and avoid bolding certain numbers in the main text.

(Minor suggestions for reference only)

---

> ### Author Response · Authors · 2024-11-20
>
> Thank you for your detailed review. We are glad that you find our analysis and approach to be informative and valuable! Below are our response to the weaknesses and questions:
>
> **[W1] Better explanation for performance degradation of PTv3 with respect to increasing window size**
>
> >We agree that the observed degradation may be attributed to limited data availability rather than the model's inherent capacity limitations. We have improved the corresponding sections on lines 234 and 298 of the updated paper to emphasize this point.
>
> **[W2] Additional 3D object detection experiment**
>
> > Following your suggestion, we provide an additional 3D object detection result on ScanNet v2 dataset:
> | backbone | mAP@0.25 |
> | -------- | ------- |
> | PTv3  | 71.3   |
> | MEEPO | 72.2 (+0.9) |
>
> **[Q1] Clarification of PTv3 experimental setup**
>
> > We conduct our experiments mainly on V100 which does not support flash attention. Based on our test, PTv3 does consume more memory in this setup.
>
> **[Q2] Possibility of changing to impactful title**
>
> > Thanks for your suggestion! We will certainly take this into consideration. However, currently we haven't settled on a suitable title yet.
>
> **[Q3,Q4] Presentation improvement suggestions**
>
> > Thanks for your suggestions! We have followed them to make the table fonts slightly larger and reduce the use of bold text.

---

### Author Response · Authors · 2024-11-27
**General Response**

Dear Reviewers and Area Chair,

We sincerely thank you for your valuable time and effort in reviewing our paper.

In this paper, we provide a comprehensive analysis of architectural properties of commonly used point cloud segmentation operators, investigate the reasons behind Mamba-based networks' under-performance in this task, and propose simple yet effective solutions to address these issues. These insights culminate in the development of a novel architecture for point cloud segmentation, named MEEPO, which achieves state-of-the-art performance. Our research offers many valuable insights for designing efficient and effective point cloud segmentation networks.

All the reviewers have acknowledged several strengths of our paper, including its **clear structure and readability (HJ9v, fEVg)**, **meaningful and thorough analysis (HJ9v, U9M8, wi14)**, and **comprehensive ablation experiments (HJ9v, wi14)**. Additionally, there is unanimous agreement regarding the **impressive performance and efficiency** of our proposed network.

Most weaknesses mentioned are minor issues related to writing presentations and clarity. In response to the constructive feedback given, we have carefully revised the manuscript, making the following key improvements:

1. Enhanced the explanation of PTv3's performance degradation with increasing window size (lines 234 and 298)
2. Added a stride=1 comparison in Table 7(e) (line 446)
3. Incorporated a Strided SSM ablation study in Table 7(d) (line 441)
4. Clarified parameter count details in Table 7(a) (lines 196 and 469)
5. Provided more detailed explanation of Mamba's point cloud processing order (line 189)
6. Expanded the discussion of Mamba's local bias (line 822 in the Appendix)
7. Included data demonstrating MEEPO's effectiveness in processing increasingly larger point sets (line 841 in the Appendix)
8. Reduced bold text usage (lines 357 and 370)
9. Corrected the Figure 5 reference (line 251)
10. Added citation to Bi-Mamba+ (line 518)

We are pleased to note that all the engaged reviewers expressed satisfaction with these revisions, awarding positive scores (8, 6, 6, 6) after their evaluation. We have also carefully addressed the weaknesses raised by reviewer HJ9v but was unable to obtain his/her response. All in all, we strongly believe that our revised paper has met the rigorous standards required for an ICLR submission.

Thank you again for your thoughtful feedback and support.

Sincerely,

The Authors

---

### Meta-Review · Area_Chair_tiu5 · 2024-12-16

**Metareview:**

The paper receives 4 positive and 1 negative rating after rebuttal. Although the paper has some merits like competitive results with faster runtime and lower memory cost, the reviewers pointed out a few critical concerns about 1) technical contributions compared to other Mamba-based approaches, and 2) results other than semantic segmentation. After taking a close look at the paper, rebuttal, and discussions, the AC agrees with reviewers' feedback, especially on the minor architectural change of the existing Mamba methods but applying to the studied task. Without further explanations about the main contributions or additional results, e.g., other point cloud tasks, it is not convincing to show enough technical novelty, and hence the AC suggests the rejection decision. The authors are encouraged to improve the paper based on the feedback for the next venue.

**Additional Comments On Reviewer Discussion:**

In the rebuttal, some of the concerns like technical clarity are addressed by the authors. However, even the discussions are not actively participated during the post-rebuttal discussion period, the AC finds that the authors have failed to provide detailed responses to most reviewers' critical questions, e.g., 1) model training with more data from reviewer U9M8, 2) tasks beyond segmentation from reviewer U9M8 and fEVg (only one additional experiment was provided), 3) main technical contributions compared to Mamba-based methods from reviewer HJ9v, wi14. XaY8, and fEVg, especially on the over-claimed Bidirectional SSM where VisionMamba already proposed. This significantly limits the technical novelty of the proposed framework, given the marginal performance improvement (also, the improved runtime and memory saving is naturally brought by Mamba anyway). Overall, the AC took a close look at all the contents and agrees that the authors have not addressed the above concerns well in the rebuttal, in which the paper still requires to be significantly improved before making it ready for publication.

---

### Decision · Program_Chairs · 2025-01-22

Reject